# Position: An Empirically Grounded Identifiability Theory Will Accelerate Self-Supervised Learning Research

**Patrik Reizinger** [1 2]   **Randall Balestriero** [3]   **David Klindt** [4]   **Wieland Brendel** [1 2 5]

## Abstract

Self-Supervised Learning (SSL) powers many current AI systems. As research interest and investment grow, the SSL design space continues to expand. The Platonic view of SSL, following the *Platonic Representation Hypothesis* (PRH), suggests that despite different methods and engineering approaches, all representations converge to the same Platonic ideal. However, this phenomenon lacks precise theoretical explanation. By synthesizing evidence from Identifiability Theory (IT), we show that the PRH can emerge in SSL. There is a gap between SSL theory and practice: Current IT cannot explain SSL's empirical success, though it has practically relevant insights. Our work formulates a blueprint for SSL research to bridge this gap: we propose expanding IT into what we term Singular Identifiability Theory (SITh), a broader theoretical framework encompassing the entire SSL pipeline. SITh would allow deeper insights into the implicit data assumptions in SSL and advance the field towards learning more interpretable and generalizable representations. We highlight three critical directions for future research: 1) training dynamics and convergence properties of SSL; 2) the impact of finite samples, batch size, and data diversity; and 3) the role of inductive biases in architecture, augmentations, initialization schemes, and optimizers.

## 1. Introduction

Self-supervised learning (SSL) drives many current AI breakthroughs in language, vision, and image-text models (Radford et al., 2018; 2021; Assran et al., 2023). This remarkable success builds on a plethora of algorithms and engineering practices. However, the model zoo makes SSL hard to navigate, divides the attention for a better theoretical understanding and, we argue, dampens future advancements. SSL is likely to continue being a foundational paradigm of machine learning, but it needs to be consolidated, similar to the program of unifying deep learning through geometry (Bronstein et al., 2021). There were recent attempts to unify the SSL model zoo (Wang & Isola, 2020; Morningstar et al., 2024; Bizeul et al., 2024; Fleissner et al., 2025). Huh et al. (2024) even argued, by positing the Platonic Representation Hypothesis (PRH), that neural networks converge to effectively the same representation, despite different setups, loss functions, data modalities, and engineering practices. However, their analysis does not provide a conclusive mathematical answer *when and why* this happens. We do have theories that can provide such an answer, but a gap exists between SSL theory and practice. Prevalent frameworks such as identifiability theory (IT) cannot explain the role of initialization, stop gradients, or learning dynamics, and neither can solve problems like dimensional collapse (Jing et al., 2022) or loss saturation. An even larger problem of current theory is generalization: The promise of self-supervised representations to work well on many downstream tasks alludes to "universality," or at least out-of-distribution (OOD) generalization. However, current theories mostly focus on the i.i.d. case, and empirical methods generally perform well only in-distribution (Montero et al., 2021; Schott et al., 2021; Montero et al., 2024; Mayilvahanan et al., 2024a;b).

[1]Max Planck Institute for Intelligent Systems, Tübingen, Germany [2]ELLIS Institute Tübingen, Germany [3]Department of Computer Science, Brown University, Rhode Island, USA [4]Cold Spring Harbor Laboratory, Cold Spring Harbor, New York, USA [5]Tübingen AI Center, Germany. Correspondence to: Patrik Reizinger <patrik.reizinger@tuebingen.mpg.de>.

*Proceedings of the 42$^{nd}$ International Conference on Machine Learning*, Vancouver, Canada. PMLR 267, 2025. Copyright 2025 by the author(s).

---

**Position:** empirical advancements alone are insufficient to accelerate SSL research, we need to bridge the gap between theory and practice.

We need an empirically grounded theory, which we call Singular Identifiability Theory (SITh), to accelerate SSL research by:
  (i) Designing realistic data generating processes (DGPs), grounded in empirical observations;
  (ii) Formalizing when and why self-supervised representations are converging;
  (iii) Providing principled recommendations for designing and evaluating SSL.

**This position paper argues that to accelerate SSL research, we need to develop a theoretical framework grounded in empirical observation.** Acknowledging the success of IT, we call for its extension to what we term Singular Identifiability Theory (SITh)—alluding to Singular Learning Theory (SLT) (Watanabe, 2009; 2020), which extended learning theory (LT) to considerations relevant to neural networks. SSL practitioners need to rely on principled guidance derived from theory, whereas theoreticians need to concentrate on studying practically relevant aspects of SSL—we provide concrete examples and research questions in Tab. 1. Our **contributions** are:
- We provide an approachable introduction to identifiability theory (IT) for practitioners, demonstrating how SSL practice can benefit from IT (§ 3);
- We call for an empirically grounded extension of identifiability, termed Singular Identifiability Theory (SITh) to close the gap between SSL theory and practice, mainly relying on the concept of a DGP (§ 4);
- We provide an extensive overview of the open questions in SSL, formulate concrete research questions and promising research directions to suggest how empirical observations and theoretical results can drive SSL forward (Tab. 1).

## 2. Background

**SSL drives most AI breakthroughs.** Long gone are the days when supervised, semi-supervised, and unsupervised learning co-existed as the main tracks of top AI conferences. The emergence of Internet-scale datasets coupled with the need for foundation models (Bommasani et al., 2021)—models capable of solving many tasks post-training—has moved the landscape of AI under one umbrella: self-supervised learning (SSL). With access to data and compute, hundreds of SSL methods have emerged (Balestriero et al., 2023). As a result, SSL now holds the state-of-the-art across numerous domains like remote sensing and medical data (Wang et al., 2022; Krishnan et al., 2022), and applications such as zero-shot reasoning (Radford et al., 2021; Zhai et al., 2023), multi-modal learning (Girdhar et al., 2022). The distinct engineering choices and implementation details make it hard to empirically test every hypothesis one may have in mind. Yet, such hypothesis testing is becoming more important than ever to understand possible robustness, fairness, compositional, or structural limitations of existing methods. Without a clear understanding of today's limitations, it is getting harder to develop tomorrow's methods.

**Self-supervised representations are often but not always similar.** Theoretical and empirical evidence indicates a surprising similarity between some self-supervised representations. Roeder et al. (2020) proved that in specific cases, two trained models learn a similar representation up to a linear representation. Works on relative representations and model stitching (Moschella et al., 2023; Cannistraci et al., 2023; Norelli et al., 2023; Fumero et al., 2024; Maiorca

et al., 2024) demonstrated that learned self-supervised representations can be related via simple (e.g., linear) transformations. A similar notion of representation linearity was demonstrated in language models (Park et al., 2023; Marconato et al., 2024; Jha et al., 2025), building from earlier observations of analogy making in word embeddings like `word2vec` (Mikolov et al., 2013; Arora et al., 2016). The PRH (Huh et al., 2024) proposed an explanation of why such convergence can happen, alluding to Plato's allegory of the cave. Huh et al. (2024) formalized three hypotheses, corresponding to the effect of: i) *function class*, ii) *data sets and loss functions*, and iii) *regularization*. Earlier work by Duan et al. (2020) suggested a link between model convergence and identifiability. We discuss evidence for their importance in SSL and highlight that current theories cannot fully explain their effect (cf. Appx. B for details). Morningstar et al. (2024) concluded that the data augmentations matter more than the SSL method for downstream classification accuracy on ImageNet (Krizhevsky et al., 2012). Ciernik et al. (2024) demonstrated that SSL representations are often, but not universally, similar (cf. Fig. 1). The prevalent problems of dimensional collapse (Jing et al., 2022) or the projector phenomenon (Jing et al., 2022; Chen et al., 2020; von Kügelgen et al., 2021), suggest that the PRH can only be true for a set of models. Thus, a mathematically conclusive answer as of *when and why* the PRH can hold (in SSL) eludes the research community. Our work aims to guide research interest towards answering these questions.

## 3. Identifiability theory

This section provides an intuitive introduction to the terminology of identifiability theory (IT) (§ 3.1), and demonstrates IT's merits for practitioners on the example of Sim-CLR (§ 3.2). We also review the state of the field (§ 3.3); and highlight many IT-driven applications (§ 3.4).

### 3.1. The terminology of identifiability

Consider a diverse collection of images like ImageNet. For each image, certain "ingredients" or latents came together to create it: the type of object, its position, color, lighting conditions, and so on. Identifiability theory (IT) provides a mathematical framework to study whether we can discover these underlying properties just by looking at the images.

Imagine these images were created by a virtual "renderer"—what IT calls a *data generating process (DGP)*—that takes these latents as input and produces the final image (Kulkarni et al., 2015). For instance, to generate a photo of a red car, this renderer would need inputs specifying "car" as the object class, "red" for its color, and values for its position. The key question IT asks is:

*Given only the images, under what conditions can we work backwards to uncover the original latents through SSL?*

This is not always straightforward since there might be mul-

tiple valid ways to describe the same latent. Take color: we could represent it using RGB values or HSV coordinates—both produce identical images (Higgins et al., 2018). IT handles this by grouping these equivalent descriptions into *equivalence classes*. To pin down a specific solution, we need additional assumptions about the DGP, e.g., regarding the distribution of latents or the function class of the renderer (Sprekeler et al., 2014).

### 3.2. What can identifiability bring to SSL? A gentle introduction with SimCLR

To illustrate how IT can improve our understanding of SSL methods, we consider SimCLR (Chen et al., 2020).

**Step 1: the birth of SimCLR.** SimCLR was proposed from an empirical perspective without any identifiability theoretic considerations. Its success inspired theoreticians to try to understand its operating principles.

**Step 2: Uniformity and alignment.** The first theoretical result by Wang & Isola (2020) conceptualized the Sim-CLR loss in terms of two opposing "forces:" an attractive force termed *alignment* pulling the representations of similar (augmented, positive) samples together, whereas a repelling force termed *uniformity* pushing dissimilar (negative) samples apart. This conceptual framework provided some practical guidance on design choices, including the temperature value or batch size. However, problems such as dimensional collapse (Jing et al., 2022) (when some latent information is not learned), or the emergence of the projector—when downstream performance is better for the non-ultimate layer—remained unexplained.

**Step 3: Identifiable DGP.** Zimmermann et al. (2021) showed—building on prior works in nonlinear Independent Component Analysis (ICA) (Hyvarinen & Morioka, 2016; Hyvarinen et al., 2019; Khemakhem et al., 2020a;b; Hyvarinen & Morioka, 2017)—that the SimCLR objective is optimized when the neural network learns to parametrize a DGP on the hypersphere, where the augmentations are modeled with a von Mises-Fisher (vMF) conditional distribution. This result showed how a few assumptions are sufficient for identifiability: a hypersphere latent space with uniformly distributed samples, augmentation distributions according to a vMF, and an invertible encoder. Furthermore, this result made the *implict* assumptions about the data and the underlying DGP explicit and pinpointed their unrealistic nature: the vMF conditional is isotropic, which does not correspond to realistic augmentations, including large crops, which are crucial for SSL. Thus, IT can model *simplified* SSL methods, but not the practically used versions.

**Step 4: towards a more realistic theory.** A later extension by Rusak et al. (2024) proved identifiability for a conditional which can model augmentations with differ-

ent strengths. Yet, we still do not understand (and cannot mitigate) dimensional collapse (Jing et al., 2022) or the projector.

### 3.3. The state of identifiability theory

The prevalent method family with identifiability guarantees is Independent Component Analysis (ICA) (Comon, 1994; Hyvärinen & Pajunen, 1999; Hyvarinen et al., 2001; Hyvärinen et al., 2023). ICA methods aim to extract (conditionally) independent latent variables, called sources—this problem is also called Blind Source Separation (BSS). Though identifiability is impossible in the nonlinear case without further assumptions (Hyvärinen & Pajunen, 1999; Locatello et al., 2019), recent nonlinear ICA methods demonstrated identifiability results for increasingly realistic settings, pushing forward our understanding of when, how, and why SSL could work (Khemakhem et al., 2020b;a; Reizinger et al., 2022; Gresele et al., 2021; Morioka et al., 2021; Morioka & Hyvarinen, 2023; Hyvarinen & Morioka, 2016; Hyvarinen et al., 2019; Klindt et al., 2021). Identifiability guarantees are also central in the field of causal inference (Pearl, 2009). Causal methods are promising data efficiency and generalization, thus, they are of interest for practice. The new field of Causal Representation Learning (CRL) (Schölkopf et al., 2021) also has identifiability guarantees (von Kügelgen et al., 2023; von Kügelgen, 2024; Wendong et al., 2023; Brehmer et al., 2022; Ahuja et al., 2022), some of which were shown to be deeply connected to ICA methods (Reizinger et al., 2024b; Hyvärinen et al., 2023). Recently, Reizinger et al. (2024a) showed that even for standard classification tasks, a simple DGP can be formulated that enjoys linear identifiability. Moreover, this is easily estimated by maximum likelihood cross-entropy minimization, generalizing the result of Roeder et al. (2020) and providing evidence about why Huh et al. (2024) observed representational similarity across many training objectives—namely, they were different formulations of cross-entropy minimization. Despite delivering insights about widely-used SSL methods (Zimmermann et al., 2021), most theoretical works are fairly idealistic as they make strong assumptions, including assuming infinite data and infinite training time (i.e., model convergence).

### 3.4. Applications: why practitioners should care

The biggest critique of IT—and machine learning theories in general—is that 1) most progress in the field comes from empirical techniques or scaling data and compute and 2) theoretical results cannot realistically be transferred to practical scenarios (Sutton, 2019). We discuss this view in detail in § 5, but emphasize that algorithms with identifiable DGPs are used in many scientific and applied domains, including: robotics (Locatello et al., 2020; Lippe et al., 2023), dynamical systems (Lippe et al., 2022; Rajendran et al., 2023), neuroimaging (Himberg et al., 2004; Hyvarinen & Morioka,

2016), neuroscience (Zhou & Wei, 2020; Schneider et al., 2023), genomics (Morioka & Hyvarinen, 2023; Morioka & Hyvärinen, 2023), structural biology (Klindt et al., 2024), ant colonies (Dingling et al., 2024), and climate science (Yao et al., 2024). What is common in IT-driven applications is the focus on inferring the *correct* (unveiling the underlying science) and *robust* (generalizing, even OOD) mechanism from the data. Indeed, recent works in IT showed that OOD generalization is possible (Brady et al., 2023; 2025; Wiedemer et al., 2023a;b). Thus, these practically relevant considerations illustrate the benefit of IT-driven SSL design.

# 4. Position: Singular Identifiability Theory as a blueprint to close the gap between SSL theory and practice

With the example of SimCLR, we showed how IT can improve our understanding of SSL methods (§ 3.2), particularly by making the assumptions on the data explicit via a DGP. However, IT relies on simplifications and cannot explain many practical phenomena, including dimensional collapse (Jing et al., 2022), or OOD behavior. ***Thus, we need to move beyond the current paradigm and address the practical realities of modern machine learning. We call for an extension of IT to what we term Singular Identifiability Theory (SITh), and detail what it can bring to SSL.*** Similarly to how Singular Learning Theory (SLT) (Watanabe, 2009; 2020) was proposed to address, among others, the reality that due to certain symmetries, (the Fisher information matrix of) neural networks can have singularities. We want to emphasize that SITh, in its current form, is an umbrella term for a *future* theory and a *blueprint* (Bronstein et al., 2021) for SSL research. We aim to provide signposts along which the path towards this new theory can be traversed. Thus, we review the gaps between SSL theory and practice (Tab. 1). We detail the empirical and/or theoretical evidence in the following subsections, pinpoint potential synergies, and formulate research questions, answering which, we believe, will move the field forward.

## 4.1. The data augmentation gap: unrealistic DGPs

Extensive evaluations demonstrate that data augmentation strategy matters more than the SSL method (Morningstar et al., 2024). As we showed with SimCLR in § 3, the DGP is a useful construct to model the augmentations—and is at the core of identifiability results in Contrastive Learning (CL) (Hyvarinen & Morioka, 2016; Morioka et al., 2021; Zimmermann et al., 2021; Rusak et al., 2024). This means that changing an augmentation should be reflected in the DGP, and such changes might make some latents less feasible or impossible to learn (von Kügelgen et al., 2021; Daunhawer et al., 2023). The problem with current IT is that the modeling assumptions in the DGP are *too simplistic*. Using a hypersphere as latent space and simple

conditionals such as vMF distributions are neither principled nor realistic—for example, they cannot capture common augmentations such as (heavy) crops (Zimmermann et al., 2021; Rusak et al., 2024; Reizinger et al., 2024a).

**Question 4.1.1.** Moutakanni et al. (2024) suggest that prior knowledge is not required for augmentation design. To what extent is this the case, does this impose performance limitations, and is it possible to develop a theoretical understanding of this phenomenon?

## 4.2. The asymptotic data gap: data set and batch size

Identifiability results assume infinite data, batch size, and converged, i.e., IT is an *asymptotic* theory—apart from Campi & Weyer (2002); Lyu & Fu (2022). In practice, data set size and data or task diversity (Elmoznino et al., 2024; Raventós et al., 2023) matters[1]. However, it is unclear whether data set size improves performance or it only correlates with properties like diversity in the sense of the ICA literature (intuitively, data coming from diverse environments), which is crucial for identifiablity (Hyvarinen & Morioka, 2016; Morioka et al., 2021; Rajendran et al., 2023; Khemakhem et al., 2020a; Reizinger et al., 2024a). More data can *seemingly* improve OOD performance. However, Mayilvahanan et al. (2024a;b) showed that the real reason is that the OOD tasks became in-distribution as they were included in the training data. Batch size is also important in practice (Chen et al., 2020; Rusak et al., 2024), but there are no corresponding theoretical results. Recently, Reizinger et al. (2024a) showed that there is a connection between InfoNCE and DIET, where one difference is calculating the loss over the whole dataset for DIET, whereas InfoNCE uses a mini-batch of data. As both methods are identifiable, this might suggest that the role of batch size can be understood theoretically.

**Question 4.2.1.** Lyu & Fu (2022) provided a finite-sample analysis on the data diversity condition in nonlinear ICA, whereas Lyu et al. (2021) analyzed the sample complexity of recovering shared (content) latents in multi-view SSL. Can we use such results to provide practically meaningful recommendations on data set size and data diversity to achieve identifiable representations empirically?

**Question 4.2.2.** Is the number of tasks a model is trained on (called "task diversity" in Raventós et al. (2023)) a causal factor in improving model performance or is it more of a proxy to diversity in the ICA sense (Hyvarinen et al., 2019), i.e., more tasks presumably mean more diverse tasks? For example, Bansal et al. (2024) provides evidence that data diversity matters.

---

[1]Though "diversity" in these contexts often refers to the number of tasks, e.g., in Raventós et al. (2023)

| Problem | Domain | Status Quo | RQ's |
|---------|--------|-----------|------|
| **Data augmentations** | Theory | ? Augmentations are modeled in the DGP, not in observation space | 4.1.1 |
| | Practice | ✓ The role of augmentations is understood | |
| **Finite data** | Theory | ✗ Finite-sample analysis is almost entirely missing from IT | 4.2.1 |
| | Practice | ✓ Scaling laws show the effect of data set size | |
| **Data diversity** | Theory | ✓ Data diversity is well-defined in the DGP view of IT | 4.2.2 |
| | Practice | ✗ Data diversity is sometimes entangled with data set size | |
| **Finite time** | Theory | ? Crude understanding of (linear) training dynamics, but not tied to IT | 4.3.1 |
| | Practice | ✓ Convergence differs across models | 4.3.2 |
| **Loss saturation** | Theory | ? IT does not consider time, learning dynamics provides insights for linear models | 4.3.2 |
| | Practice | ✗ It is unclear why convergence speeds differ | |
| **Inductive biases** | Theory | ✗ Results missing or not reflecting practical choices | 4.4.1 |
| | Practice | ✓ The role of architecture is well-understood | |
| **Initialization** | Theory | ✗ IT does not consider initialization | 4.4.1 |
| | Practice | ✓ It is well understood what initialization are useful | |
| **Dimensional collapse** | Theory | ✓ The DGP view defines what latents collapse | 4.5.1 |
| | Practice | ? Computational tricks and regularizers might help collapse | |
| **Projector** | Theory | ✗ No explicit result, some settings reflect a linear projector | 4.5.1 |
| | Practice | ? The projector is not eliminated, at most ameliorated | |
| **Compositionality** | Theory | ✓ Theoretical results exist for compositional generalization | 4.6.1 |
| | Practice | ✗ Empirical methods struggle to generalize compositionally | |
| **CL/non-CL** | Theory | ✗ Theory for non-CL methods is missing | 4.7.1 |
| | Practice | ✓ Representations are similar | 4.7.2 |
| **Evaluation** | Theory | ✓ IT characterizes what latents are learned | 4.8.1 |
| | Practice | ✗ Mostly benchmark-related, principles often missing | 4.8.2 |

*Table 1.* **The gaps between SSL theory and practice:** RQ is a shorthand for research question, ✓ denotes a comprehensive understanding; ? incomplete or related results; and ✗ mostly uncharted territory.

## 4.3. The finite time gap: learning dynamics and loss saturation

Identifiability theory cannot distinguish between the convergence speed of models, as it focuses on models at the global optimum of the loss, i.e., at convergence. As experiments demonstrate, not achieving convergence to the global optimum—e.g., by a saturating loss, when it is very close to the optimum, but small differences can lead to qualitatively different models (Liu et al., 2023; Reizinger et al., 2024c)—is often the barrier to high-quality representations (Simon et al., 2023; Rusak et al., 2024; von Kügelgen et al., 2021; Kadkhodaie et al., 2024; Roeder et al., 2020). Another example of this is *grokking*, where the training loss saturates long before the true, generalizing solution is discovered (Power et al., 2022). Recently, theoretical insights started to emerge that provide some guidance on how to improve latent recovery via augmentation design, ensemble models, or hard negative sampling (Eastwood et al., 2023; Rusak et al.,

2024), which might guide further research understanding how such choices affect learning dynamics (potentially in terms of a DGP). Current IT aims to recover all latent factors, ignoring the minimality argument of Achille & Soatto (2018), and the practical considerations of efficiency and compression. "Partial" identifiability results such as those of von Kügelgen et al. (2021); Daunhawer et al. (2023) are generally seen in negative light—even though some latent factors might not matter downstream.

**Question 4.3.1.** Why is it the case that sometimes, even though identifiability is only provable for some latent factors, with a large enough latent space and training time, the other latent factors can also be learned (to some extent) (von Kügelgen et al., 2021)?

**Question 4.3.2.** Can we understand what model components, such as vanishing initialization (Simon et al., 2023; Kunin et al., 2024; Draganov et al., 2025), determine the convergence speed differences of SSL methods and how

data augmentation design and negative sampling change learning dynamics?

### 4.4. The architecture gap: teacher/student networks, initialization, stop gradient

The SSL model zoo hinges on particular architectural choices like stop gradients and predictor networks, mostly in the non-contrastive domain (Bardes et al., 2021; Zbontar et al., 2021; Oquab et al., 2024). The necessity of engineering practices and model components needs to be critically evaluated to improve robustness, decrease computational cost, and to make theoretical analysis easier—leading to a potentially wider range of insights. The DIET (Ibrahim et al., 2024) instance discrimination method was proposed as a simplified SSL pipeline to avoid the theoretically not well-understood components of SSL (e.g., stop gradients). DIET performs comparably to other SSL methods and is proven to identify a cluster-centric DGP (Juhos et al., 2024; Reizinger et al., 2024a). These works demonstrated that SSL can provably work without many computational tricks, hopefully helping practitioners to design more efficient pipelines in the future.

**Question 4.4.1.** Morningstar et al. (2024) showed that many different algorithmic knobs do not substantially alter the downstream performance of self-supervised representations. How can we use this observation to develop theoretical results showing what are the necessary and sufficient architectural and data components to prove identifiability?

### 4.5. The projector gap: dimensional collapse

The projector phenomenon refers to better latent recovery/downstream performance at the other-than-the-ultimate layers of the trained network (Chen et al., 2020). That is, some dimensions are collapsed (Jing et al., 2022). Note that this is distinct from *neural collapse* (Papyan et al., 2020) where the model learns one-hot predictions of the training data rather than learning the correct conditional probabilities as assumed, e.g., in Reizinger et al. (2024a). The projector is seen as "wasting" resources by training the last (few) layers, and its mitigation is actively researched (Tian, 2022; Bordes et al., 2023; Jing et al., 2022; Song et al., 2023; Bizeul et al., 2024; Xue et al., 2024; Saunshi et al., 2021). The projector is an accidental phenomenon—i.e., no one designs an SSL pipeline by saying "the last $k$ layers are the projector, i.e., I expect them to be useless for downstream performance." What is a deliberate choice is to use a linear or a nonlinear projector. Interestingly, a nonlinear projector is not necessarily better—see comparison in, e.g., (Mialon et al., 2022, Fig. 2). We argue that there is a possible theoretical explanation of both 1) why the projector phenomenon occurs, and 2) why a linear projector is often better than a nonlinear one. Roeder et al. (2020); Hyvarinen & Morioka (2016); Reizinger et al. (2024a) assumed that the trained neural network is a composition of a nonlinear encoder $\mathbf{f}$

and a linear projection matrix $\mathbf{W}$, yielding $\mathbf{W} \circ \mathbf{f}$. Then they prove identifiability of $\mathbf{z} = \mathbf{f}(\mathbf{x})$ in a log-linear model, where the $\mathbf{z}$ is fed through a softmax to yield the loss. We hypothesize that, in certain cases, this log-linear model can describe and explain the (linear) projector's occurrence. So far, this potential link between identifiability results and the projector phenomenon has eluded the community, and requires further research.

**Question 4.5.1.** Roeder et al. (2020); Reizinger et al. (2024a) show that under a specific DGP, the key construct for identifiability is a log-linear model, i.e., estimating cross entropy with a softmax. Does this provably explain the (linear) projector phenomenon? Is it possible to develop an identifiability result without the projector?

### 4.6. The generalization gap: OOD and compositionality

Identifiability results have recently been extended to specific OOD scenarios in computer vision, based on compositionality (Brady et al., 2023; 2025; Wiedemer et al., 2023b;a; Lachapelle et al., 2023). These results exploit that composition of attributes and objects is often straightforward to model in a DGP. Their proofs rely on imposing structural constraints such as additivity on the generative model. Most empirical representation learning methods do not generalize compositionally (Montero et al., 2021; 2024; Schott et al., 2021; Wiedemer et al., 2023a), emphasizing the need for theoretical insights to design better SSL methods. Interestingly, in the language domain, both the first theoretical results (Ahuja & Mansouri, 2024; Xie et al., 2022), and empirical techniques and observations (Bansal et al., 2022; Ruoss et al., 2023; Mészáros et al., 2024) exist showing that OOD generalization might be possible, at least in terms of length generalization or rule extrapolation.

**Question 4.6.1.** Is it possible to provide OOD or compositional generalization results across data modalities, such as in image-text models? If yes, is there a qualitative or quantitative difference w.r.t. the inductive bias of each modality?

### 4.7. The Platonic representation gap: the contrastive and non-contrastive dichotomy

A prevalent categorization of SSL is the contrastive/non-contrastive dichotomy. While categorizations like these are undoubtedly valuable, they might obscure the overall trends within the SSL model landscape. However, recent initiatives are striving to bring unity to the field of SSL (Balestriero et al., 2023; Morningstar et al., 2024; Cabannes et al., 2023). Garrido et al. (2022) showed that both method families contrast some properties; for this reason, they propose the names *sample-contrastive* (for CL) and *dimension-contrastive* (for non-contrastive). Balestriero & LeCun (2022) analyzed contrastive and non-contrastive paradigms with a spectral embedding approach, showing that such methods correspond to specific local and global spectral embedding methods (e.g., SimCLR was shown to

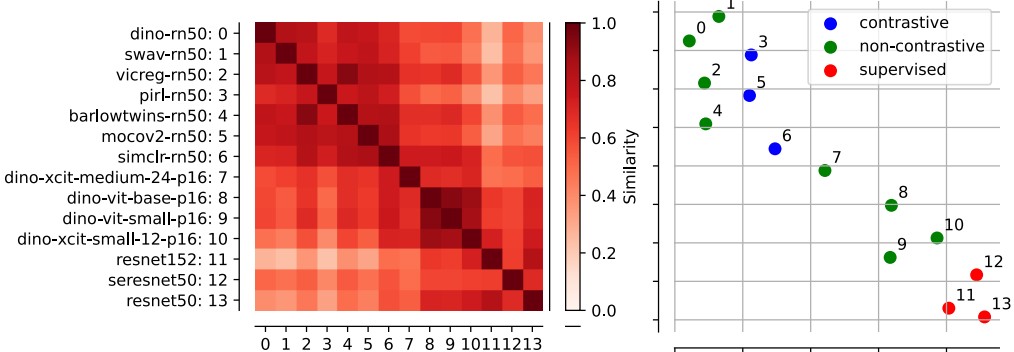

*Figure 1.* **Representational similarities between contrastive and non-contrastive methods trained on ImageNet-1k: (Left):** the similarities are calculated with an RBF-kernel–based CKA with $\sigma = 0.4$. We include supervised baselines for reference. Reproduced from (Ciernik et al., 2024) with the authors' permission. **(Right):** a two-dimensional UMAP embedding of similarity matrix colored by method types (contrastive, non-contrastive and supervised).

be equivalent with ISOMAP (Tenenbaum et al., 2000) under specific assumptions). Similarly, t-SNE can be made contrastive with minimal modifications (Böhm et al., 2022). There is also evidence that both paradigms relate to entropy and mutual information estimation (Liu et al., 2022). Wang & Isola (2020) interpreted InfoNCE as an interaction between uniformity (i.e., entropy estimation), and alignment (i.e., invariance w.r.t. augmentations). The InfoNCE family is shown to minimize cross-entropy (Zimmermann et al., 2021; Rusak et al., 2024); the same holds for the (non-contrastive) instance-discrimination method called DIET, which is also related to InfoNCE (Ibrahim et al., 2024; Reizinger et al., 2024a). These principles are also present in non-contrastive methods, though usually via regularizers. For example, VICReg (Bardes et al., 2021) also estimates a lower bound on entropy (Shwartz-Ziv et al., 2022). One distinction is the absence of an explicit DGP in most non-CL methods. That is, the DGP is mostly defined implicitly, with the exception of (Reizinger et al., 2024a).

> **The contrastive and non-contrastive split posits a sterile dichotomy**
>
> Contrastive and non-contrastive methods are driven by the same principles, thus, we need to consider their similarities, and not focus on their differences as they represent different means to the same end.

Experimental evidence demonstrates that (identifiable) contrastive and non-contrastive representations fare similarly on some downstream task (Morningstar et al., 2024) and that their representations can be highly similar (Ciernik et al., 2024)—cf. Fig. 1 and also note similarity is not universal, e.g., ViT-based models are less similar. However, identifiability guarantees for non-contrastive methods are mostly

missing—with the exception of DIET (Ibrahim et al., 2024; Reizinger et al., 2024a). The empirical similarity suggests that some non-contrastive methods might be identifiable.

**Question 4.7.1.** Based on the similarity of contrastive and non-contrastive representations (Morningstar et al., 2024; Ciernik et al., 2024) and the identifiability of contrastive methods, can we prove identifiability for non-contrastive methods?

**Question 4.7.2.** The representations of contrastive and non-contrastive methods are often similar (Ciernik et al., 2024) (cf. Fig. 1), though their convergence speed is highly distinct (Simon et al., 2023). Can we theoretically understand in which (data, model size, etc.) regimes which method is expected to perform best?

### 4.8. The evaluation gap: going beyond ImageNet

Most SSL methods evaluate performance on ImageNet (Krizhevsky et al., 2012) downstream classification, without acknowledging that such an evaluation can only falsify claims about classification-related latents. For example, if orientation is not required for classification, then a more universal representation (e.g., one capturing orientation) will not perform better—though this makes a difference in terms of "universality" of the representation. Indeed, SSL methods capturing more latents sometimes have (slightly) lower downstream classification accuracy (Rusak et al., 2024), which might suggest their infeasibility—though that depends on what they will be used for. *We need to question the practice of measuring the "universality" of representation with a single scalar on a highly specialized task.* A good example of a more comprehensive evaluation is DINOv2 (Oquab et al., 2024). Another gap, especially for evaluating identifiability claims, is the lack of large-scale data sets with latent information—proxies such as

ImageNet-X (Idrissi et al., 2022) exist, though they only have binary labels. Recent works started using aggregate statistics (e.g., rank conditions) to evaluate the representation quality (Agrawal et al., 2022; Garrido et al., 2023; Thilak et al., 2023). These methods seem to be somewhat predictive of even OOD performance, we lack a theoretical understanding when and why this is the case.

> **Evaluating self-supervised representations requires more than ImageNet classification**
>
> SITh can improve SSL evaluations by using a DGP to focus on the relevant latent factors and design more principled benchmarks.

**Question 4.8.1.** How can we evaluate the "universality" of self-supervised representations for robustness? I.e., how should we move away from downstream classification on ImageNet? What are the *set* of downstream tasks we need to evaluate SSL methods?

**Question 4.8.2.** How can we develop ImageNet-scale (synthetic) datasets with latent information to evaluate the identifiability of self-supervised representations? Can we use rendering pipelines akin to DisLib (Locatello et al., 2019) to develop more principled benchmarks?

### 4.9. Position summary: We need an empirically grounded identifiability theory for SSL

To improve our understanding of the (implicit) operating principles of SSL methods such as SimCLR (Zimmermann et al., 2021) (§ 3.2). Recent IT results also improved our understanding of compositional generalization (Brady et al., 2023; 2025; Wiedemer et al., 2023b;a; Lachapelle et al., 2023). Yet, current IT cannot fully explain empirical observations in SSL. To quote Nobel-prize–winning physicist Richard P. Feynman: *"It doesn't matter how beautiful your theory is, if it doesn't agree with the experiment, it's wrong."* IT models the data via an underlying data generating process (DGP), however, the design of the DGP does not exploit guidance from empirical observations, which leads to presumably the biggest critique of identifiability theory: *Identifiability guarantees are not valuable per se, only to the extent they can describe empirical phenomena, or—and only time will tell this—they aid the development of such theories that explain such phenomena.*

When a theorist constructs a DGP, the focus should not only be on identifiability but also on the match with reality (Klindt et al., 2021). Vice versa, when a practitioner sets up a pre-text task for SSL, they should think about the implicit DGP that they might be assuming and turn to theory to ask whether this setup results in an identifiable DGP (and estimation procedure). Constructing a DGP on the hypersphere is common in theory (Zimmermann et al., 2021; Reizinger et al., 2024a). However, it is unclear how well

this actually corresponds to a realistic DGP (for images). Analogously, building a pre-text around data augmentations is an empirically successful construct, however, it will only result in theoretic identifiability if the data augmentations cover all degrees of freedom in the data—a rather unrealistic assumption, mirrored by findings on the importance of the right augmentations (Chen et al., 2020; Oquab et al., 2024; Morningstar et al., 2024).

## 5. Alternative Views

Recent years' breakthroughs in AI were often fueled by scaling up models, compute, and data (Sutton, 2019). Though empirical algorithmic improvements are also crucial for breakthroughs like the recent DeepSeek-R1 (DeepSeek-AI et al., 2025) reasoning model. The scaling-related observations were documented in so-called scaling laws (Hernandez et al., 2021; Zhai et al., 2022; Henighan et al., 2020; Hestness et al., 2017; Kaplan et al., 2020), analyzing trends in loss reduction. Especially in the case of Large Language Models (LLMs), scaling data and compute is still significant for progress, maintaining the popularity of this opinion in the community in general.

**Counterarguments in SSL:** Realistically, engineering efforts are crucial to solve the problems of real-world SSL systems. However, we are seeing the SSL community shift beyond scaling.[2] After scraping the majority of data from the Internet, and pushing the boundaries of compute beyond previously unimaginable limits, we need a new approach. Sorscher et al. (2022) theoretically showed that scaling laws can sometimes be overcome via pruning. More data can also imply that OOD tasks become in-distribution (Mayilvahanan et al., 2024a;b), qualitatively changing what we as a community thought about the true role of scaling data. We propose an identifiability theoretical framework as an alternative to the scaling view, still prevalent in LLM research. We believe that having a more principled approach towards SSL, which incorporates the empirical observations, is the way to move SSL forward. Particularly, improving OOD generalization performance in SSL is hard to do empirically. First, most empirical representation learning methods, even despite aiming to learn disentangled representations, do not generalize compositionally (Montero et al., 2021; 2024; Schott et al., 2021; Wiedemer et al., 2023a). Second, it is hard to get the intuition without using a model like a DGP. We believe that Singular Identifiability Theory (SITh) can help overcome these limitations, and lead to more efficient, principled, and robust SSL methods.

## 6. Conclusion

Our position paper hit a critical note to highlight what we conceive as a big barrier before closing the gap between

---

[2] Personal observations at the NeurIPS 2024 Workshop: Self-Supervised Learning - Theory and Practice

SSL theory and practice. We argued that thinking in terms of an empirically-grounded data generating process (DGP) can accelerate SSL research. We formulated our suggestion in terms of an extension to identifiability theory, called Singular Identifiability Theory (SITh), to foster synergies between the two subfields. To provide a common starting point, we included an intuitive introduction of identifiability for practitioners (§ 3), demonstrating how SSL practice can benefit from IT. However, more research needs to be done, illustrated by the many open questions in both theory and practice (Tab. 1). As we are close to hitting the limits of scaling-driven advances, we need to turn towards more principled, theoretically grounded approaches. This is especially important when machine learning models are deployed in safety-critical environments such as autonomous driving or healthcare. We believe that SITh facilitates asking better research questions, many of which we shared, to drive SSL forward by improving how we design, evaluate, and understand SSL algorithms.

## Acknowledgements

The authors would like to thank Laure Ciernik, Evgenia Rusak, Roland Zimmermann, Julius von Kügelgen, Michael Kirchhof, Lucas Maes, Vishaal Udandarao, Ameya Prabhu, Karsten Roth, Omar Chehab, Runtian Zhai, Kamilė Stankevičiūtė, Thaddäus Wiedemer, Pradeep Ravikumar, Luigi Gresele, Quentin Garrido, Alice Bizeul, Mark Ibrahim, and Attila Juhos for their valuable suggestions for shaping the manuscript. The authors thank the International Max Planck Research School for Intelligent Systems (IMPRS-IS) for supporting Patrik Reizinger. Patrik Reizinger acknowledges his membership in the European Laboratory for Learning and Intelligent Systems (ELLIS) PhD program. This work was supported by the German Federal Ministry of Education and Research (BMBF): Tübingen AI Center, FKZ: 01IS18039A. Wieland Brendel acknowledges financial support via an Emmy Noether Grant funded by the German Research Foundation (DFG) under grant no. BR 6382/1-1 and via the Open Philantropy Foundation funded by the Good Ventures Foundation. Wieland Brendel is a member of the Machine Learning Cluster of Excellence, EXC number 2064/1 – Project number 390727645. This research utilized compute resources at the Tübingen Machine Learning Cloud, DFG FKZ INST 37/1057-1 FUGG.

## Impact Statement

This paper presents work whose goal is to advance the field of Machine Learning. There are many potential societal consequences of our work, none which we feel must be specifically highlighted here.

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

## A. On absolute vs relative identifiability

We need to tackle one nuance when talking about identifiability and the alignment of representations: namely, what do we mean by identifiability? There are two notions in the literature, which we will call *relative* and *absolute identifiability*. These are defined as follows:

**Definition A.1** (Absolute identifiability of the ground-truth representation). Given a ground-truth probabilistic model $p_\theta$, and a learned family of probabilistic models parametrized by a neural network at the optimum of its training loss $p_{\hat\theta} : \hat\theta \in \hat\Theta$, the inferred model identifies the ground-truth representation if $\forall \hat\theta \in \hat\Theta$ and a particular equivalence relationship $\sim_{abs}$ it holds that $p_{\hat\theta} \sim_{abs} p_\theta$.

That is, absolute identifiability means that there is an equivalence relationship (e.g., permutation or scaling) between the learned (family of) representations and the ground-truth one. "Family" refers to, e.g., training the same model with different seeds. This identifiability notion is more prevalent in the ICA literature (Hyvärinen & Pajunen, 1999; Hyvärinen, 2013; Hyvärinen et al., 2023).

On the other hand, relative identifiability posits an equivalence relationship between two trained models, i.e., it does not make any claims about the ground-truth representation.

**Definition A.2** (Relative identifiability of a pair of learned representations). Given a pair of learned probabilistic model families $p_{\theta_1}, p_{\theta_2} : \theta_1 \in \Theta_1, \theta_2 \in \Theta_2$ parametrized by (not necessarily the same) neural networks at the optimum of their respective training losses, they are said to be identified in a relative sense if there exists an equivalence relationship $\sim_{rel}$ such that $\forall \theta_1 \in \Theta_1, \theta_2 \in \Theta_2 : p_{\theta_1} \sim_{rel} p_{\theta_2}$.

## B. The Platonic Representation Hypothesis revisited

Huh et al. (2024) argued that the i) function class, ii) dataset, and iii) regularization, determine the convergence of representations, which are important in SSL (Cabannes et al., 2023; Balestriero et al., 2023; Morningstar et al., 2024). By synthesizing evidence from the identifiability literature, we showcase the mathematical results answering some of these questions. However, we will also show the negative results and the limits of current IT, enabling us to precisely specify future research directions. Before that, we emphasize that the PRH's claims are about the similarity of two *learned* representations (such as in Roeder et al. (2020)), whereas most identifiability results are about an *absolute* sense between a learned model and the assumed underlying DGP—we discuss the differences in Appx. A.

**Function class and model capacity.** Restricting the function class is a conventional technique to prove identifiability (Hyvärinen & Pajunen, 1999; Locatello et al., 2019; Bloem-Reddy & Teh, 2020; Buchholz et al., 2022). That is, in some cases no restriction can imply non-identifiability, indicated by the impossibility results of Locatello et al. (2019); Hyvärinen & Pajunen (1999). A restriction of the functions class in form of a log-linear model was also shown to be key for recent identifiability results in (self-)supervised learning (Hyvarinen & Morioka, 2016; Hyvarinen et al., 2019; Roeder et al., 2020; Reizinger et al., 2024a; Shimizu et al., 2006; Hoyer et al., 2008; Lachapelle et al., 2020; Gresele et al., 2021). Compositional generalizationalso hinges on particular (mainly additive) decoder structures (Lachapelle et al., 2023; Brady et al., 2023; Wiedemer et al., 2023a;b; Brady et al., 2025; Mahajan et al., 2024; Ahuja et al., 2022) Such results indicate that the PRH's validity is potentially restricted to a specific model class. The investigation of Ciernik et al. (2024) also suggests a varying degree of similarity even between self-supervised models (Fig. 1).

**Data set and training objective.** Many self-supervised models optimize an objective that corresponds to the cross entropy between the distributions of the underlying and the learned DGPs (Zimmermann et al., 2021; Huh et al., 2024; Reizinger et al., 2024a). This implies that, e.g., changing the way positive pairs are generated implies adapting the loss function to avoid a mismatch (Zimmermann et al., 2021; Rusak et al., 2024). von Kügelgen et al. (2021) proved that when some latent factors are not changed by the augmentations, then only those are provably identifiable—others collapse, as observed in the projector phenomenon (Chen et al., 2020; Rusak et al., 2024; Jing et al., 2022). This supports empirical observations about the importance of augmentations (Morningstar et al., 2024), and suggests that different sets of augmentations can have different "Platonic ideals".

**Regularization.** Huh et al. (2024) posited that deep networks are biased towards learning simple functions (Nakkiran et al., 2019). In a discussion about the non-identifiability of LLMs, Reizinger et al. (2024c) noted that the theoretical understanding of a wide range of inductive biases is still missing—this applies to SSL in general, too—at least for inductive biases beyond the ones expressed via the loss function, the function class, or data augmentations. Rusak et al. (2024) suggested an anisotropic conditional can worsen the learning dynamics of high-variance factors, though more work is required to precisely understand nonlinear learning dynamics in SSL (Simon et al., 2023).

**Summary.** In our view, the PRH made a first important step formalizing the similarities of learned representations. However, it does not provide a clear theoretical framework to precisely investigate such questions

## C. Acronyms

**BSS** Blind Source Separation

**CL** Contrastive Learning
**CRL** Causal Representation Learning

**DGP** data generating process

**i.i.d.** independent and identically distributed
**ICA** Independent Component Analysis
**IT** identifiability theory

**LLM** Large Language Model

**LT** learning theory

**OOD** out-of-distribution

**PRH** Platonic Representation Hypothesis

**SITh** Singular Identifiability Theory
**SLT** Singular Learning Theory
**SSL** self-supervised learning

**vMF** von Mises-Fisher

