# OpenReview forum: "Position: An Empirically Grounded Identifiability Theory Will Accelerate Self Supervised Learning Research"
_ICML.cc/2025/Position_Paper_Track — ICML 2025 Position Paper Track poster_

### Official Review · Reviewer_ZnYz · 2025-03-07

**Significance:** 3
**Argument Clarity:** 2
**Rating:** 2
**Confidence:** 4

**Questions:**

1. In multiple parts of the text the problem of latent collapse is mentioned and the paper of Papyan et al.,
2020 is cited. I suppose the authors refer to the collapse or dimension collapse in the case of SSL. The paper of Papyan et al. is focused on neural collapse which is actually a desired property of networks that leads to an almost opposite effect to dimension collapse as the centers of classes lie on a Simplex ETF, thus they span efficiently the dimensions of the latent space. Is this a mistake or is there some other intention?

**Discussion Potential:**

2

**Paper Summary:**

The paper's position is that bridging the gap between empirical results and theory in Self-Supervised Learning (SSL) using tools from Identifiability Theory (IT) could accelerate research in the field.

Towards this direction the authors suggest the Singular Identifiability Theory (SITh) which is composed of a list of 12 current problems/weaknesses of SSL and their current status together with some questions, which if answered, would yield progress in the corresponding problems. Some highlights are questions on contranstive vs non-contrastive learning, convergence, on the issues of latent collapse and the existence of projectors, and on data and convergence properties.

## update after rebuttal

After the discussion with the authors, I have updated my original score to weak reject. The reason is that the paper does have some value due to review of recent developments in the field. I am still leaning though towards rejection because of introduction of the misleading concept of "Singular Identifiability Theory" which, in my opinion, is not really a well-defined theory but rather an ad hoc list of improvement suggestions.

**Position:**

Yes

**Position In Title:**

Yes

**Related Work:**

3

**Strengths And Weaknesses:**

Strengths:

1. The paper puts together a descent collection of recent works and developments on SSL and on the attempts to generate theoretical insights on it using IT.

Weaknesses:

1. The paper lacks direction and narrative. Its goal is to support the position that a theory based on IT would help SSL, but to do so, it only lists several works in that direction along with related questions and their current status. No concrete insights or a mathematical framework is provided to put those together. One would expect for example something in the lines of the group actions and layers which preserve them which is introduced in the geometric deep learning paper by Bronstein et al., 2021. It is actually stated in the introduction that a program along those lines would be needed in SSL. Instead, the current work feels more like a plain list of papers.

2. It also lacks content. It mentions a lot of papers and their results but doesn't get into any details, e.g. some formulas or images which would explain the strengths of those methods and how they fit together and in the suggested roadmap.

3. The name of the suggested framework is very confusing. The term "Singular" from Singular Learning Theory refers very explicitly to the singularities of algebraic varieties used to analyze loss landscapes which singularities are in turn a fundamental feature of algebraic geometry. Using this very specific term in this name only to sound similar to another field is not a great choice.

Suggestions:

1. Though my opinion is that the work is not fit for a position paper, I acknowledge the effort of putting together the recent advances in the field. If one would add more details on each method and highlight more the relations between them, then maybe it could be re-submitted as a review paper.

2. In the bibliography, there are a lot of citations of arxiv papers which have actually been also published on conferences. I would suggest checking and updating those.

**Support:**

2

---

> ### Author Rebuttal · Authors · 2025-04-01
>
> We thank **Reviewer ZnYz** for the detailed consideration of our manuscript. We address the raised concerns below.
> First, we would like to highlight that **Reviewer oFd4 stated that our argument is excellent, Reviewers oKBL, and pt49 highlighted the clarity of our contributions, Reviewer pt49 praised the potential impact and vision, Reviewer oKBL found our methodology promising, whereas Reviewers oKBL and pt49 praised our effort to identify the gaps in the literature**.
>
> ## Direction and narrative
> The Reviewer mentioned that our paper lacks concrete insights. First, we point out that **Reviewers pt49 and oKBL** highlighted the clarity of our contributions and the impactfulness of our vision. We highlight that **we provide concrete examples of how particular methods can be improved** - in our Tab. 1 and the case studies in Secs. 3.2, 4.5, 4.7. To help the reader navigate our proposal and to translate our ideas into concrete results, we will add more concrete examples of how existing results can be utilized to lead to new contributions - for such examples, please [refer to our response to Reviewer pt49](https://openreview.net/forum?id=ET6qJpllEi&noteId=yeUU8BP8wi).
>
> We would also like to highlight that **the purpose of a position paper is not to focus on a particular method**. From the call for position papers: _“Examples include (but are not limited to) an argument in favor of or against a particular research direction (not a particular algorithm or solution), a call to action...“_
> The reviewer mentioned that our submission lacks an analogous statement as the advocation for groups actions in (Bronstein et al., 2021). We respectfully disagree. Our submission **clearly states these tools, including finite-sample analysis, training dynamics, compositionality, and inductive biases** (particularly architecture, data diversity, and initialization).
>
> ## Content
> If we understand the Reviewer's comment, they criticize the depth, not the content itself (if we misunderstood the argument, we ask the reviewer to kindly clarify it). We draw the Reviewer’s attention to **two in-depth analyses we already discuss in our paper**:
> 1. We made an in-depth case for identifiability in Sec 3.2, using SimCLR to showcase how insights since its inception improved the field.
> 2. We provided an in-depth discussion of the Platonic Representation Hypothesis in App. B, including Fig. 1 with empirical evidence (and a novel analysis), and discussed its three components and their connection (and related current insights) in identifiability theory
>
> ## Neural collapse
> Thank you for pointing out this discrepancy; this is indeed our mistake, which we will correct in our updated manuscript. We will also clarify that in the context of SSL, any type of collapse is unwanted.
>
> ## Singular Identifiability Theory
> We thank the reviewer for raising their concern regarding the name choice. We believe that **the singularity of the Fisher Information matrix**, which can come about in deep neural networks due to overparametrization, parameter symmetries, or flat loss landscapes, **captures a crucial missing aspect of current identifiability theories**.
> Thus, to signify a similar shift required for learning theory, we believe _this name choice reflects our main message._
> We also believe the same mathematical toolbox used for SLT can help advance identifiability theory. We will ensure that the reasoning behind the name is more precise in our paper.
> Nonetheless, we are open to suggestions from the reviewer for a better name.

---

> > ### Comment · Reviewer_ZnYz · 2025-04-08
> >
> > I do agree that the significance of the of the topic, I have adjusted the corresponding ratings.
> >
> > I still believe though that the paper does not introduce a concrete research direction, but feels more like a review of the progress of using IT in explaining SSL phenomena. A title like e.g. "Position: Bridge the gap between identifiability theory and practice in SSL" would describe the paper contents much better and then the list presented in table 1 would define this gap. Wouldn't give a name to a theory which is not described at all in the paper.

---

> > > ### Author Response · Authors · 2025-04-09
> > >
> > > Thank you for reconsidering your scores!
> > >
> > > **Could you please provide what would be a concrete research direction in your opinion?**
> > >
> > > We believe that we provided concrete examples beyond reviewing IT in the field of SSL, for example:
> > > - The concrete research questions: these are detailed enough such that a new graduate student in the field could start working on them
> > > - The exact theoretical tools needed to advance IT in SSL: finite sample analysis, training dynamics, and overparametrization.
> > >
> > > We suspect that the Reviewer means something different by "concrete," so we would like to ask you to clarify what you meant so that we can improve our submission. Thanks in advance!

---

### Official Review · Reviewer_pt49 · 2025-03-09

**Significance:** 2
**Argument Clarity:** 2
**Rating:** 3
**Confidence:** 4

**Questions:**

Formalizing SITh. Could the authors detail how SITh might be formally developed? Would it extend
existing nonlinear ICA approaches with finite-sample considerations, or introduce entirely new theoretical
tools inspired by singular learning theory?
Clarifying Causal Inference vs. Latent Causal Discovery. Causal inference typically concerns
identifying causal effects among observed variables with known structures, whereas latent causal discovery
explicitly targets identifying latent causal variables and structures from observational data. Could authors
clarify the intended connection between SITh and these distinct causal concepts?
OOD Generalization and Causality. The authors briefly discuss OOD generalization and causal links
to identifiability. Could they elaborate on how SITh specifically integrates causal insights or OOD robustness
explicitly within its framework?

[In line 224] Correct typos such as ’identifiablity’ (should be ’identifiability’).
[In line 80] Standardize terminology consistently as ’self-supervised learning.’

**Discussion Potential:**

2

**Paper Summary:**

This position paper argues that current theories inadequately explain the empirical success and representa￾tional similarities observed across self-supervised learning (SSL) methods. The authors propose extending classical Identifiability Theory (IT) into a new framework, Singular Identifiability Theory (SITh), designed to encompass realistic SSL scenarios and bridge theory with practical observations. They clearly outline gaps between theory and practice, provide research questions to guide future developments, and advocate SITh as a path forward for SSL research

## Update after rebuttal:
I thank the authors for their responses. I maintain my score and recommend weak accept.

**Position:**

Yes

**Position In Title:**

Yes

**Related Work:**

3

**Strengths And Weaknesses:**

Important Problem. The paper addresses the timely and critical need for a unified theory of SSL. It effectively highlights the Platonic Representation Hypothesis (PRH), noting current analyses do not suﬀiciently
explain when and why SSL methods converge to similar representations.
Novel Framework and Clear Contributions. The contributions are clearly articulated: (1) providing
an accessible introduction to Identifiability Theory (IT), illustrated concretely via SimCLR; (2) proposing
the novel Singular Identifiability Theory (SITh), a framework extending IT to encompass data augmentation,
optimization dynamics, and other practical considerations; and (3) offering a structured roadmap of research
questions and challenges (Table 1) to guide the SSL community.
Effective Synthesis of Theory and Empirical Findings. The authors skillfully connect existing empirical SSL insights (e.g., representational similarities, augmentation importance) to theoretical frameworks.
The SimCLR example, which shows SSL objectives can reflect identifiable generative models, reinforces their
conceptual synthesis.
Identification of Critical Gaps. The paper systematically identifies important gaps between theory and
practice—covering finite samples, data augmentations, training dynamics, architecture, and representation
collapse. These gaps are well-supported by references and are structured effectively through clear research
questions.
Impactful Vision. The vision of SITh is compelling: to provide SSL with theoretical rigor similar to that
which identifiability provided for ICA, enhancing interpretability, robustness, and generalization. Its explicit
emphasis on empirical grounding and neural network-specific challenges makes the proposal both original
and practically relevant.


Lack of Clarity in Key Concepts. Certain central concepts like PRH and SITh could benefit from
clearer explanations. PRH’s three determining factors (function class, data distribution, regularization)
should be explicitly defined and linked to identifiability. Similarly, SITh is currently abstract and could
appear as merely a conceptual umbrella without concrete formalisms. Clarifying precisely what distinguishes
SITh from classical IT, including how the term ”Singular” relates to neural network-specific issues, would
strengthen the paper.
Justification of Scope and Identifiability Assumption. The paper implicitly positions identifiability
theory as central to SSL without deeply justifying why alternative frameworks (e.g., information-theoretic
or purely empirical approaches) might not suﬀice. Given identifiability’s stringent assumptions, the authors
could further clarify conditions under which identifiability remains realistic for complex, high-dimensional
data.
Limited Depth in Some Discussions. While the breadth of identified gaps showcases complexity,
discussions sometimes remain superficial due to the broad scope. Each topic (e.g., training dynamics or
architecture heuristics) merits deeper exploration. A more focused analysis on fewer gaps could strengthen
the narrative’s coherence.

**Support:**

2

---

> ### Author Rebuttal · Authors · 2025-04-01
>
> We thank **Reviewer pt49** for their detailed review, highlighting the importance and clarity in our problem setting and contribution, our effective identification of the gaps in the field, and deeming our vision impactful. We are also grateful for the suggestions and questions, which we address below.
>
> ## Depth of Discussions
> We agree that the topics merit deeper exploration, this was our goal to communicate. Our understanding of the call for papers is to focus on the position, not necessarily to dive deep in any particular direction.
> We believe that the presented gaps hinder progress; our goal was to provide a comprehensive case - we are receptive to the reviewer’s opinion about what to emphasize more.
>
> ## PRH details
> Thank you for your constructive suggestions. We **detailed the 3 components of the PRH in App. B** and highlighted that current identifiability theory only utilizes assumptions on the data distribution and the function class to prove identifiability - c.f. also [(Xi and Bloem-Reddy, 2023)](http://arxiv.org/abs/2206.00801).
>
>
>
> ## Singular Identifiability Theory
>
> ### SITh vs IT
> We listed concrete components constituting SITh in Tab. 1, but we will clarify the distinction between SITh and classical IT, and how the term “singular” relates to issues in neural networks.
> **SITh vs IT: the the role of overparametrization, parameter symmetries, flat loss landscapes are missing from current IT, whereas SITh does incorporate them.** For details regarding the term "singular," please refer to our [response to Reviewer oFd4](https://openreview.net/forum?id=ET6qJpllEi&noteId=ZSgXTV6BaQ)
>
> ### Could the authors detail how SITh might be formally developed?
> - Making assumptions more realistic and reflecting reality: current works omit practical design choices in their assumptions (e.g, using CNNs or Transformers). **Example**: identifiability log-linear models with a linear projector and a nonlinear backbone can be extended to include properties of the backbone (Reizinger et al., 2025).
> - Addressing realistic effects (finite batch, time) by characterizing how probable identifiability is (i.e., if we have an identifiability guarantee, will SGD converge to that optimum. If yes when and what is the probability of this convergence?) **Example:** learning dynamics analysis (Simon et al., 2023) could be utilized to provide finite-time identifiability results.
> - Developing new performance metrics that measure the robustness/”universality” of the representation (instead of, e.g., downstream classification). **Example:** to improve the measurement of robustness, the field of SSL could inspire from the field of causal inference, where evaluations under (sparse) distribution shifts are prevalent to measure whether the algorithm extracted robust causal mechanisms or only spurious features (Perry et al., 2022).
>
> ## Scope and Assumptions
> We would like to clarify our view: we _do not say that other frameworks are inferior or inappropriate_. We use the existence of other frameworks to highlight the connections between seemingly different SSL methods.
>
> From identifiability, we emphasize the **central role of a data-generating process** (DGP), which acts as a world model. This is the most important aspect that even practitioners can adopt (e.g., for augmentation design), without reaching for theoretical tools.
>
> Some identifiability results rely on restrictive assumptions. However, recent advances rely on fairly realistic scenarios, e.g. (Roeder et al., 2020) use the standard classification setup, whereas (Zimmermann et al., 2021) that of SimCLR. These proofs hold for widely used methods. However, a lot of work needs to be done (e.g., finite-sample analyses).
>
> ## Extension or new theoretical tools?
> We believe that new theoretical tools are needed as current ICA theory does not address many practical concerns (e.g., initialization, dynamics), for which SLT can serve as the inspiration. We will clarify this point in our submission.
>
> ## Clarification: SITh, causality, OOD
> We reference causal identifiability results, though we did not explicitly make the connection to causality. As causality is concerned with OOD generalization (via interventional and counterfactual queries), there is a connection to causal discovery and representation learning, i.e., when either causal structure and/or causal variables need to be learned from observational data
>
> **Compositionality can extend IT to OOD cases** - such works assume that e.g. images are composed of independent objects. Intuitively, compositionality provides information beyond the support of the training data (e.g., by positing that all combinations of two attributes are possible even if all pairs are not in the training set).
> This compositional approach is inherent in causal works, too, via the causal Markov factorization of the joint data distribution. This suggests that by formulating the SSL task in terms of these causal mechanisms, identifiability guarantees can be extended.

---

> > ### Comment · Reviewer_pt49 · 2025-04-01
> >
> > Thank you for addressing my concerns. I will keep my score.

---

### Official Review · Reviewer_oKBL · 2025-03-13

**Significance:** 3
**Argument Clarity:** 3
**Rating:** 3
**Confidence:** 3

**Questions:**

Please refer to the weakness in the above section

**Discussion Potential:**

3

**Paper Summary:**

The paper advocates for the systematic use of identifiability theory, introduced by seminal work on ICA and extensively expanded, particularly in the area of causality in ML, to improve understanding and advance the field of self-supervised learning (SSL). Specifically, the methodology involves specifying a ground truth data-generating process (typically expressed as a probabilistic graphical model) and analyzing the conditions under which inference can successfully recover the underlying latent factors. \
The paper proposes several axes of investigation, including:
1. Improving the design of data-generating processes to bridge the gap between identifiability theory and empirical observations in SSL.
2. Considering finite data regimes, thus extending beyond existing asymptotic results.
3. Studying the dynamics of learning, complementing results obtained at optimality.
4. Enhancing the understanding of heuristics for architecture design.
5. Investigating the problem of collapses.
6. Examining generalization through the lens of compositionality.
7. Unifying existing classes of self-supervised methods, particularly contrastive and non-contrastive approaches.
8. Proposing new evaluation benchmarks beyond ImageNet.

**Position:**

Yes

**Position In Title:**

Yes

**Related Work:**

2

**Strengths And Weaknesses:**

**Strengths**
- The paper is clearly written, with a well-articulated position and several directions for future investigation.
- The work identifies important challenges and provides preliminary evidence to support some of its claims.
- More broadly, it seeks to bridge the communities of causality in ML and self-supervised learning, which could foster fruitful discussions.
- Overall, I find the methodology of specifying a data-generating process to study the identifiability of representation learning strategies (including self-supervised learning) to be a promising approach.

**Weaknesses & Suggestions for Improvement** \
While the paper presents a compelling perspective, there are aspects that could be refined to improve its soundness, depth, and connection to prior work:
- On Collapses (Points 4 & 5): The discussion on collapses is imprecise and could be strengthened. The paper equates dimensionality collapse with neural collapse [Papyan et al., 2020], but these concepts are distinct. Dimensional collapse occurs when the latent representation lies on a lower-dimensional subspace, which does not necessarily imply the collapse of individual latents. Neural collapse, on the other hand, refers to the clustering of classifier outputs at the extrema of the probability simplex. Expanding the discussion to incorporate alternative perspectives on collapses would improve the paper’s completeness, particularly in light of recent work exploring failure modes and corresponding theoretical frameworks [1, 2].
- On the Unification of SSL (Point 7): The paper presents a narrow perspective on unification, omitting key literature and making speculative claims. Prior work has already used probabilistic graphical models to unify contrastive, feature decorrelation, and clustering-based SSL approaches under a Bayesian modelling umbrella [3]. Additionally, recent research has sought to bridge contrastive learning with other unsupervised learning strategies [4]. The claims that "the contrastive and non-contrastive split posits a *sterile* dichotomy" and “contrastive and non-contrastive methods are driven by the same principles” are intriguing but would benefit from theoretical substantiation. Are there specific non-contrastive methods that exhibit contrastive properties? Or are all non-contrastive approaches contrastive ? Could the converse be true ? Addressing this point would clarify the argument and strengthen the paper’s position.

**Reference** \
[1] Sansone. The Triad of Failure Modes and A Possible Way Out. NeurIPS Workshop 2023 \
[2] Sansone, Lebailly, Tuytelaars. Failure-Proof Non-contrastive Self-Supervised Learning. arXiv 2024 \
[3] Sansone, Manhaeve. GEDI: GEnerative and DIscriminative Training for Self-Supervised Learning. arXiv 2022 \
[4] Alshammari, Feldmann, Hershey, Freeman, Hamilton. I-CON: A Unifying Framework for Representation Learning. ICLR 2025

**Support:**

2

---

> ### Author Rebuttal · Authors · 2025-04-01
>
> We thank **Reviewer oKBL** for their constructive feedback, the provided additional references, and acknowledging that our position is well-articulated and identifies clear challenges in the field of SSL.
> We address the Reviewer’s remarks and questions below.
>
> ## On Collapses (Points 4 & 5)
> Yes, thanks for pointing this out, and also for sharing additional references, which we will incorporate in our paper. We **agree with the reviewer that detailing the different collapse phenomena (dimensional, cluster, intracluster)  is crucial to have a more complete overview of the field**. We have made a mistake and will correct this in our updated submission. _You are right in pointing out that neural collapse and dimensionality collapse are distinct phenomena_. We will make it clear that we are referring to the former. In fact, there is an **interesting relationship between neural collapse and finite data identifiability theory** that we would like to highlight. Neural collapse occurs when cross-entropy training on finite data fails, and the model does not learn the correct posterior probabilities with class uncertainties but collapses on the one-hot training labels. This also _breaks a key assumption in identifiability theory_, which assumes that the global optimum of the cross-entropy loss on infinite data has been achieved, i.e., where the model learns the correct posterior probabilities with class uncertainty. Thus, understanding under what conditions (training duration, dataset size) neural collapse occurs may be instrumental also to get a better handle on finite, i.e., non-asymptotic identifiability results that often make this crucial assumption.
>
> ## On the Unification of SSL (Point 7)
> Thank you for providing these additional references, which we will incorporate in our paper! Importantly, **we only use the contrastive-non-contrastive framework to pinpoint a shortcoming of the current SSL literature, we do not say that this is the single or the best framework to categorize SSL**. Thus, we will make it clearer why we put more emphasis on the contrastive/non-contrastive categorization (to highlight its potential shortcomings) and will include a broader discussion on other unifying frameworks in SSL. We already reference [information theoretic](https://arxiv.org/abs/2210.11464) and [probabilistic](http://arxiv.org/abs/2402.01399) frameworks. These other frameworks also suggest that the contrastive/non-contrastive divide is problematic, as both categories fit into a single one in said other frameworks.
>
>
> ## Are there specific non-contrastive methods that exhibit contrastive properties? Or are all non-contrastive approaches contrastive? Could the converse is true?
>
> This is a very important question, which pinpoints an _unfortunate terminology of the field_ - we will clarify this in our paper.
> Most importantly, we argue that based on the literature, abandoning the segmentation of SSL into contrastive and non-contrastive methods would simplify analysis.
>
> This unification is substantiated by the following evidence:
> 1. There are frameworks for SSL, as also pointed out by the Reviewer, that encompass both contrastive and non-contrastive methods.
> 2. [(Garrido et al., 2022)](http://arxiv.org/abs/2206.02574) showed that **both contrastive and non-contrastive methods contrast some properties of the data** (features vs dimensions).
> 3. That is, in some sense, all non-contrastive methods are dimension-contrastive, whereas all contrastive methods are sample-contrastive (i.e.,  contrastive methods are non-contrastive w.r.t. any other property of the data, e.g., they are not contrasting the dimensions)

---

> > ### Comment · Reviewer_oKBL · 2025-04-07
> >
> > Thank you for the answers and the clarifications. Overall, I'm positive about the work. Can you please suggest concrete modifications to the text ?

---

### Official Review · Reviewer_oFd4 · 2025-03-13

**Significance:** 3
**Argument Clarity:** 2
**Rating:** 3
**Confidence:** 3

**Questions:**

See above.

**Discussion Potential:**

3

**Paper Summary:**

This paper describes open questions in self-supervised learning. In particular, it focuses on how to bridge theory with practice. It claims that these questions should be addressed in a research program called "singular identifiability theory" for self-supervised learning.

**Position:**

Yes

**Position In Title:**

Yes

**Related Work:**

3

**Strengths And Weaknesses:**

The paper is very interesting, but the position is not so clearly spelled out. According to section 3, identifiability theory basically provides a form of "ground truth" to assess whether an SSL method is better than another one. The authors give an excellent argument, saying that ImageNet classification (which is often used) is not a good metric for assessing SSL, and new metrics should be developed. Presumably, these new metrics, together with a theoretical and empirical understanding of SLL are what the authors call "singular identifiability theory". Then the authors describe a set of research questions relevant to the theory and practice of SSL. In the appendix, the authors explain the difference between relative and absolute identifiability, and mention the platonic representation hypothesis and some of its limitations.

However, the authors never define singular identifiability theory. Presumably it's a theory that answers all (or most) of the research questions in section 4, but why is it one theory? why is it called "singular identifiability theory"? why do we expect to achieve anything of that form? I think there is a bit of context missing.

**Support:**

3

---

> ### Author Rebuttal · Authors · 2025-04-01
>
> We thank **Reviewer oFd4** for their constructive feedback and their acknowledgement of our arguments. Below, we detail  the concept of Singular Identifiability Theory (SITh) and the evidence and arguments for why SITh can advance the field of SSL.
>
>
> ## Why is it one theory?
> We positioned our proposal as one theory with a **single goal: to formalize practically relevant aspects of deep learning and leverage them to develop theoretical guarantees**. We think of our proposal as _akin to the program proposed for geometric deep learning by [(Bronstein et al., 2021)](http://arxiv.org/abs/2104.13478)_
> We acknowledge that encompassing diverse aspects such as learning dynamics, initialization, and inductive biases makes this theory rather diverse. Our goal was to emphasize that these distinct aspects of SSL are all crucial for understanding the success and limitations of SSL.  We realize the naming might be misleading--thus, we are open to suggestions on how to make our point clearer.
>
>
> ## Why is it called "singular identifiability theory"?
> In our submission, we mention that **the name alludes to Singular Learning Theory (SLT) to emphasize similar incentives**, i.e., incorporating real-world effects into the idealistic settings of theory. The word **“singular” in SLT refers to the singularity of the Fisher information matrix, which is also prevalent in deep neural networks**.
> This (approximate) singularity is due to overparametrization, improper initialization schemes, parameter symmetries, or flat loss landscapes. Thus, we call our framework singular identifiability theory to refer to such practical concerns that need to be addressed in theoretical works - but current identifiability theory does not model these phenomena. However, without incorporating such practical phenomena, theoretical conclusions have limited applicability to real-world scenarios.
>
>
> ## Why do we expect to achieve anything of that form?
> That’s an excellent question! Our reasoning is as follows:
> 1. Identifiability theory has a track record of improving practical applications in AI4Science, computational neuroscience, and causal inference, as we detail it in our Sec 3.4 with 16 references.
> 2. Recent advances [(Roeder et al., 2020)](http://arxiv.org/abs/2007.00810),  [(Reizinger et al.. 2025)](https://openreview.net/forum?id=hrqNOxpItr) showed that identifiability theory could answer why log-linear models (a nonlinear backbone with a linear projection on top) work successfully for supervised and self-supervised learning.
> 3. However, the current identifiability theory has a limited scope as it does not account for many practical design choices (e.g., initialization, parameter symmetries), leading to seemingly controversial situations in SSL (e.g., SimCLR is identifiable but in practice, dimensionality collapse/the projector phenomenon occurs).
> 4. The SLT approach shifts the focus to addressing these real-world phenomena
> 5. Thus, if we incorporate practical aspects of modern deep learning into a Singular Identifiability Theory, we should be able to develop a better understanding as we do know from practice that these aspects  (e.g. initialization, parameter symmetries) do matter
> 6. **In summary:** the practice of SSL provides strong evidence for what affects the “usefulness” of the representation, which should also be reflected in theory - to answer why and how these choices matter.

---

### Decision · Program_Chairs · 2025-04-30

**Decision:**

Accept (poster)

**Comment:**

3 out of 4 reviewers suggest acceptance. The one reviewer suggesting rejection is concerned that the paper does not suggest a concrete research direction, which may be needed for a position paper. However, this AC feels that given that the position is arguing for the need for a new theory, being more precise about the nature of this theory may be difficult to do in a position paper. Further given the relative new-ness of this venue, it is counterproductive to keep too restrictive a definition of a position paper. Therefore this AC recommends acceptance.

That said, reviewers have asked for concrete textual changes to address their concerns. The authors are encouraged to follow through on the same.